

# Home sick: impacts of migratory beekeeping on honey bee (*Apis mellifera*) pests, pathogens, and colony size

Samantha A. Alger[1], P. Alexander Burnham[1], Zachary S. Lamas[2], Alison K. Brody[1] and Leif L. Richardson[3,4]

[1] Department of Biology, University of Vermont, Burlington, VT, United States of America
[2] Department of Entomology, University of Maryland, College Park, MD, United States of America
[3] Rubenstein School of Environment and Natural Resources, University of Vermont, Burlington, VT, United States of America
[4] Gund Institute for Environment, University of Vermont, Burlington, VT, United States of America

Corresponding author
Samantha A. Alger, salger@uvm.edu

## ABSTRACT

Honey bees are important pollinators of agricultural crops and the dramatic losses of honey bee colonies have risen to a level of international concern. Potential contributors to such losses include pesticide exposure, lack of floral resources and parasites and pathogens. The damaging effects of all of these may be exacerbated by apicultural practices. To meet the pollination demand of US crops, bees are transported to areas of high pollination demand throughout the year. Compared to stationary colonies, risk of parasitism and infectious disease may be greater for migratory bees than those that remain in a single location, although this has not been experimentally established. Here, we conducted a manipulative experiment to test whether viral pathogen and parasite loads increase as a result of colonies being transported for pollination of a major US crop, California almonds. We also tested if they subsequently transmit those diseases to stationary colonies upon return to their home apiaries. Colonies started with equivalent numbers of bees, however migratory colonies returned with fewer bees compared to stationary colonies and this difference remained one month later. Migratory colonies returned with higher black queen cell virus loads than stationary colonies, but loads were similar between groups one month later. Colonies exposed to migratory bees experienced a greater increase of deformed wing virus prevalence and load compared to the isolated group. The three groups had similar infestations of *Varroa* mites upon return of the migratory colonies. However, one month later, mite loads in migratory colonies were significantly lower compared to the other groups, possibly because of lower number of host bees. Our study demonstrates that migratory pollination practices has varying health effects for honey bee colonies. Further research is necessary to clarify how migratory pollination practices influence the disease dynamics of honey bee diseases we describe here.

## INTRODUCTION

Animal-mediated pollination, provided primarily by bees, is required for the production of 75% of agricultural food crops (*Klein et al., 2007*) and provides an estimated annual value of $200 billion worldwide (*Gallai et al., 2009*). Managed honey bees (*Apis mellifera*) are the most important commercially available pollinator and contribute approximately $17 billion in pollination services revenue annually to the United States (US) alone (*Calderone, 2012*). However, for over a decade, honey bees have experienced elevated colony losses (*Neumann & Carreck, 2010*; *Potts et al., 2010*; *Van der Zee et al., 2012*; *Van der Zee et al., 2013*; *Kulhanek et al., 2017*) attributed to multiple threats including pesticide exposure (*Tsvetkov et al., 2017*; *Woodcock et al., 2017*), forage availability (*Decourtye, Mader & Desneux, 2010*), and numerous pests and pathogens (*VanEngelsdorp & Meixner, 2010*). The numerous threats affecting honeybees may be exacerbated by practices inherent to the apicultural industry and required for large-scale crop pollination, specifically migratory beekeeping (*Royce & Rossignol, 1990*; *Traynor et al., 2016a*).

To meet the pollination demand of a variety of US agricultural crops, large numbers of bees are moved among crops at regional and national scales. Conditions for migratory colonies vary greatly depending on the distance traveled and the crops visited. In the most extreme cases, colonies are transported by truck to a series of monoculture crops including blueberries, cranberries, almonds, and citrus (*VanEngelsdorp et al., 2013*) for months at a time. At each stop along the journey, millions of bees from different origins converge on a single crop for the duration of bloom, which typically lasts approximately one month and may offer little forage diversity (*Decourtye, Mader & Desneux, 2010*; *Colwell et al., 2017*). Nectar, comprised of sugars and amino acids, is required to fuel flight and feed the colony while pollen, high in protein and fats, provisions developing brood (*Brodschneider & Crailsheim, 2010*). To ensure survival en route or when crops are not in bloom, colonies may be supplemented with sucrose syrup and artificial pollen, temporary but poor substitutes for the diverse array of nectar and pollen types bees obtain in natural landscapes (*Huang, 2012*). Thus, compared to their stationary counterparts, migratory colonies experience greater stress (*Simone-Finstrom et al., 2016*), greater exposure to pesticides (*Mullin et al., 2010*; *Traynor et al., 2016a*), and lower quality forage (*Brodschneider & Crailsheim, 2010*; *Decourtye, Mader & Desneux, 2010*; *Colwell et al., 2017*), all of which may increase susceptibility to disease (*Di Pasquale et al., 2013*; *Sánchez-Bayo et al., 2016*). It is well known that stress from long distance travel results in heightened bacterial and viral infections in vertebrate livestock (*Yates, 1982*). However, despite the importance of large-scale pollination events for agriculture, few studies have examined how migratory conditions may contribute to disease incidence in bees (*Welch et al., 2009*; *Zhu, Zhou & Huang, 2014*; *Traynor et al., 2016b*).

In the US, there are an estimated 2.62 million commercial honey bee colonies of which over half are contracted for crop pollination (*USDA National Agricultural Statistics Service, 2017b*). California almond pollination is the largest annual event for the migratory beekeeping industry, requiring over 1.5 million honey bee colonies (*USDA National Agricultural Statistics Service, 2017a*). It is the largest convergence of honey bee colonies in

the US, providing conditions in which pathogens are likely to be introduced, transmitted, and subsequently spread as colonies move along their human-imposed migration route (*Bakonyi et al., 2002*; *Welch et al., 2009*; *Runckel et al., 2011*; *Goulson et al., 2015*). Each acre of almonds requires an average of two honey bee colonies (*Carman, 2011*) and as bees will forage 3 km from their colonies (*Visscher & Seeley, 1982*; *Beekman & Ratnieks, 2000*; *Couvillon et al., 2015*), bees in large orchards could theoretically share flowers with bees from nearly 56,000 other colonies. While almond flowers may produce a large quantity of nectar and pollen, there is evidence that it is relatively low quality (and possibly toxic) forage for honey bees (*London-Shafir, Shafir & Eisikowitch, 2003*; *Kevan & Ebert, 2005*); moreover, the vast fields provide little forage diversity for bees and are heavily sprayed with pesticides (*California Department of Pesticide Regulation, 2016*), exposing bees to additional stress.

The spread of the most devastating honey bee parasites and pathogens has mainly occurred as a result of transporting honey bees long distances. For example, the *Varroa* mite (*Varroa destructor*), an ectoparasite and known vector of numerous RNA viruses, became a major contributor to colony losses in both North America and Europe after its introduction from Asia (*Rosenkranz, Aumeier & Ziegelmann, 2010*; *Nazzi et al., 2012*). *Nosema ceranae*, a microsporidian implicated in high colony mortality in Spain (*Higes et al., 2008*), has also reached high frequencies since its introduction from Asia to the Americas and Europe (*Klee et al., 2007*; *Chen et al., 2008*). Despite the role of long-distance travel in disease spread, there is a surprising lack of studies examining the role of migratory beekeeping in disease spread.

A limited number of observational surveys have compared disease loads of colonies belonging to migratory and stationary operations and found a higher prevalence of some pathogens in migratory colonies (*Traynor et al., 2016b*) including *Nosema ceranae* (*Zhu, Zhou & Huang, 2014*) and RNA viruses (*Welch et al., 2009*), some of which were not previously described in honey bees (*Runckel et al., 2011*). However, the focus of previous studies has been the collection of baseline disease data to characterize diseases in migratory colonies and, as such, rarely control for migratory conditions, management practices, and sampling times, all of which can significantly affect disease loads and colony health (*Runckel et al., 2011*; *Glenny et al., 2017*). Furthermore, studies examining the impact of migratory conditions on bees rarely include a control group of stationary colonies for comparison (but see *Zhu, Zhou & Huang, 2014*; *Simone-Finstrom et al., 2016*). Although migratory honey bee colonies are implicated as disease sources and could serve to introduce disease to local stationary honey bee colonies (*Welch et al., 2009*) we are unaware of previous studies that explicitly test the role of migratory colonies in the spread of diseases or parasites. Here, we conducted a two-pronged experiment in which we controlled for migratory conditions, sampling time, and beekeeper management practices. We first tested the effects of migration on honey bee colony population size, *Varroa* mite parasites, and pathogens including *Nosema* (a microsporidian) and three RNA viruses: black queen cell virus (BQCV), deformed wing virus (DWV), and Israeli acute paralysis virus (IAPV). We examined differences in the parasite and pathogen prevalence and load as well as colony size of migratory and stationary colonies. Second, we examined if there is evidence for

the transmission of diseases from migratory colonies to stationary colonies. If migration exposes bees to stressors that increase disease susceptibility, we predicted that migratory colonies would have greater pathogen prevalence and loads when compared to their stationary counterparts, and that pathogen loads in sympatric stationary colonies would increase after foraging alongside the migratory colonies for one month.

## MATERIALS AND METHODS

In February 2017, we selected 48 colonies from a North Carolina apiary that is used for the production of products (honey, colonies, etc.) rather than pollination services, and assigned each to one of the following groups: *migratory* ($n = 16$), *isolated stationary* (*isolated*) ($n = 16$), and *exposed stationary* (*exposed*) ($n = 16$; Fig. 1). We transported colonies in the migratory group from Whiteville (Columbus County), North Carolina to Coalinga (Fresno County), California (36°21′N, 120°12′W) to pollinate almonds for the duration of the bloom (approximately one month). They were then transported back to North Carolina. As typical of migratory beekeeping practices, the migratory colonies were covered by netting during transport (to reduce escapees) and temporarily brought to a nearby holding yard in California before and after pollinating almond orchards. The isolated stationary group remained in North Carolina (34°22′N, 78°36′W) and outside the flight distance from returning migratory colonies for the entirety of the experiment. To maintain similar colony densities at the isolated stationary and migratory yards, there were an additional 15 stationary colonies in the isolated yard. These colonies originated from the same North Carolina apiary and were not tested as part of the experiment.

At the start of the experiment in February 2017, all colonies had 7–9 frames of bees, and 7–8 frames with brood. To measure bee population size, we counted frames of adult bees (FOB) by assessing the coverage of adult bees on each frame and summing the estimates for all frames in the brood chamber (the lower hive body containing the queen and brood) (*DeGrandi-Hoffman et al., 2016*). Frames with brood were assessed by counting the total number of frames containing 30% capped brood. Each colony was provided a new queen by replacement with open-mated Italian (*A. mellifera ligustica*)/ Carniolan (*A. mellifera carnica*) queens in summer 2016. Colonies were matched in triplicate by frames of bees and frames of brood and randomly assigned a treatment group (migratory, isolated stationary, or exposed stationary) to ensure equal distribution across groups. Prior to the start of the experiment, in October 2016, we treated all colonies for *Varroa* mites with fluvalinate, a synthetic pyrethroid commonly used as an acaricide in honey bee colonies. No other mite or pathogen treatments were used for the duration of the experiment. To ensure that colonies would persist for the duration of the experiment, we provided supplemental feed to all colonies (in all treatment groups) on two occasions: pollen substitute prior to shipping the migratory colonies to California and upon return, 5 lbs. of fondant (sucrose and water stabilized with gelatin). As colonies grew during the duration of the study, additional hive bodies were added as needed to prevent swarming.

We compared bee population size and disease loads in the migratory and isolated stationary group three times: before the migratory group departed for California (Jan.
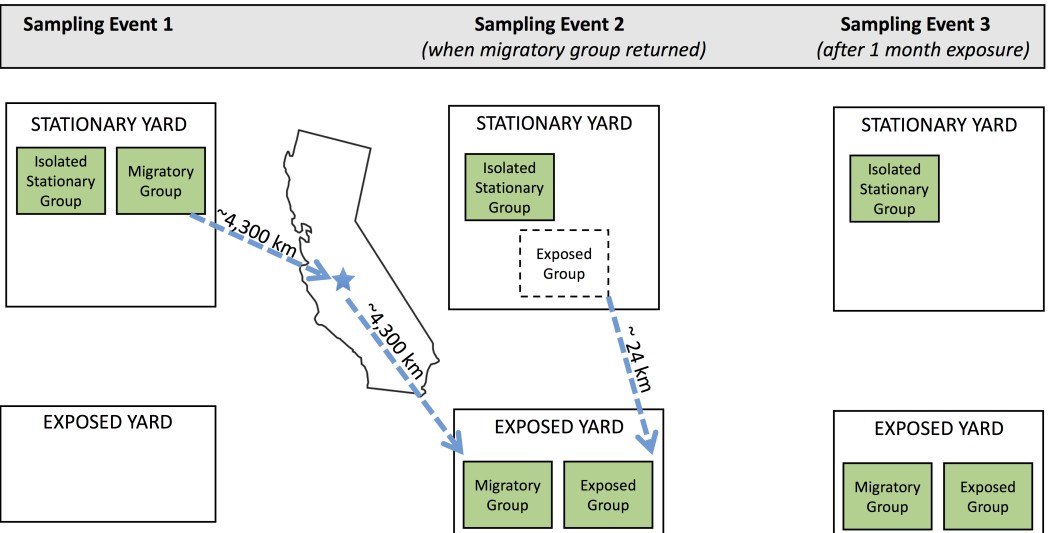

**Figure 1** **Schematic of experimental design.** Three sampling events occurred during the experiment. Three experimental groups (isolated stationary group, migratory group, and exposed group) were located in two separate apiaries in North Carolina throughout the experiment: the stationary yard (where all groups begin and the isolated stationary group remained for the duration of the experiment) and the exposed yard (where the exposed group was exposed to the migratory group). Dotted arrows show movement of colonies throughout the experiment. Between sampling events one and two, the migratory colonies were transported to California for almond pollination and back. Exposed colonies began in the stationary yard and were transferred to the exposed yard prior to sampling event two. Geographic distance between yards are specified in kilometers.

25), immediately after the migratory group returned to North Carolina from California (Feb. 28), and one month later (March 25). To test for disease spread from the migratory colonies to their stationary counterparts, we monitored the third group of colonies, the exposed stationary group, which remained in North Carolina but shared a yard with the migratory colonies once they returned from California (34°11′N, 78°46′W). We assessed bee population size and disease loads in the exposed stationary group twice: once before sharing a yard with the migratory group (Feb. 28), and again approximately one month after residing with the migratory colonies in the same yard (March 25). Land cover surrounding each of the North Carolina yarding areas were dominated by crops, mixed forest, and woody wetlands, and we expect that colonies in the two sites had similar access to early spring floral resources. Hives were housed on private land and permission was granted by the owners.

At each sampling event, we inspected all colonies for brood diseases, measured colony size, and collected bees for pathogen analyses. To estimate colony size, we measured frames of bees (FOB) as before (*DeGrandi-Hoffman et al., 2016*). We also recorded the queen status of each colony (queen-right, queenless, queen cells present, or drone-laying queen). We collected live bees from the brood chamber to detect and quantify the following parasites and pathogens: *Varroa, Nosema,* BQCV, DWV, and IAPV. To quantify *Varroa* and *Nosema* spp., we collected approximately 300 bees from the brood chamber and transferred them

to ethanol. To quantify virus prevalence and load, we collected an additional 150 bees from the brood chamber. These samples were stored and shipped to Vermont on dry ice and transferred to −80 °C for storage prior to analysis.

To examine differences in climate and weather conditions experienced by the migratory and stationary groups, we used publicly available NOAA local climatology data collected by weather stations nearest to our field sites (NOAA National Centers for Environmental Information) (Table S1).

### *Varroa* mite and *Nosema* spp. quantification

To calculate the number of *Varroa* mites per 100 bees, ethanol samples were agitated for 60 s, strained through hardware cloth to separate the mites from the bees, and all mites and bees were counted (*Lee et al., 2010*). We conducted spore counts to quantify *Nosema* spp. Although our methods did not differentiate between the two species of *Nosema, (N. apis and N. ceranae)* previous work has found *N. ceranae* to be the predominant species in many regions (*Klee et al., 2007*; *Chen et al., 2008*; *Williams et al., 2008*; *Williams et al., 2014*). To conduct spore counts, we transferred 100 bees from the ethanol sample to a plastic bag and pulverized them using a pestle on the outside of the bag for 90 s. We then added 100 mL of distilled water, allowed it to settle for 45 s, and transferred 10 μL onto a haemocytometer counting chamber. We counted spores for each sample twice under 40× magnification, averaged them, and converted to spores/bee (*Fries et al., 2013*).

### Virus quantification

To quantify BQCV, DWV and IAPV, we transferred 50 honey bees/sample on liquid nitrogen and homogenized them in an extraction bag with 10 mL of GITC buffer using protocols established by USDA-ARS Bee Research Lab Beltsville, MD (*Evans, 2006*). We followed EZNA Plant RNA Standard Protocols (Omega Bio-Tek, Norcross, GA, USA) with 100 μL of the resulting homogenate thereafter. Using a spectrometer (NanoDrop; Thermo Scientific, Waltham, MA, USA), we assessed all RNA quantity and quality and diluted all RNA extractions to 20 ng/μL prior to virus assays.

For reverse transcription of RNA and absolute quantification, we performed duplicate reverse transcription quantitative polymerase chain reaction (RT-qPCR) for each sample with a SYBR green one-step RT-qPCR kit in 10 μL reactions using the following thermal cycling program: 10 min at 50 °C (RT) followed by 1 min at 95 °C, and 40 amplification cycles of 95 °C for 15 s, 60 °C for 60 s. Lastly, we obtained the melt-curve starting at 65–95 °C (0.5 °C increments, each 2 s). We used primers specific to the positive strand of the following RNA virus targets: BQCV, DWV and IAPV, and a housekeeping gene (Actin) as a positive control of RNA extraction efficiency (Table S2). We calculated quantification using duplicate standard curves of gBlocks Gene Fragments (Integrated DNA Technologies; Data S1) that were developed using double-stranded, sequence verified genomic blocks consisting of the four targets of interest separated by ten random base pairs. Sequences of random base pairs consisting of at least 50% G and Cs were used at the beginning and terminal ends of the fragment. Efficiencies were 95.21% (BQCV), 91.06% (DWV), 90.27% (IAPV), and 90.12% (Actin), with correlation coefficients ($R^2$) ranging from 0.993–0.999.

To verify RT-PCR analyses, sequences with lengths of 100–130 bps were generated through DNA sequencing performed in the Vermont Integrative Genomics Resource using a 3130xl Genetic Analyzer.

## Data reporting

We use "pathogen prevalence" to refer to the percentage of colonies positive for a pathogen (*Varroa, Nosema,* BQCV and DWV). In addition to presence/absence data, we investigated the severity of infection by quantifying each pathogen—we refer to this as "pathogen load". Virus load (BQCV and DWV) results for each colony are presented in average virus genome copies/bee. We did not detect IAPV in our experimental colonies and it was therefore excluded from further analysis. We report *Varroa* as the number of mites per 100 bees and *Nosema* as average number of spores/bee.

## Data analysis and statistics

Before analyzing, we checked all response variables for normality using Shapiro–Wilk tests. To improve normality, *Varroa* and *Nosema* loads as well as BQCV and DWV loads (genome copies per bee) were log + 1 transformed. To establish that there were no differences between treatment groups at the outset, we analyzed all variables at the initial time step using ANOVAs for continuous variables (FOB, load of *Varroa, Nosema,* BQCV, and DWV) and chi-square tests of independence for binary variables (prevalence of *Varroa, Nosema,* BQCV, and DWV*)*.

To test whether the full suite of response variables collectively predicted colony treatment membership, we conducted classification analyses for Experiments 1 (*migratory* vs. *stationary*) and 2 (*exposed* vs. *isolated*) using linear combinations based on all response variables (except BQCV prevalence as it was fixed at 100% prevalence for all groups and as such caused model fitting failures). To examine how groups differed after experimental manipulation, we used data from sampling events two and three for Experiments 1 and 2, respectively. The models were trained using a conservative cross validation approach to reduce over-fitting the model to our data. We tested for differences between groups' centroids in multivariate space for each time point with PERMANOVA, a non-parametric MANOVA, using Euclidian distance-based dissimilarity matrices. To visualize between-group separation, the centered values from linear discriminate functions (LD1 and LD2) were plotted for each colony.

To test the effect of treatment and time on prevalence, we analyzed all pathogens (*Nosema, Varroa,* BQCV, and DWV) using separate generalized linear mixed effects models (GLMMs) using the binomial (link ="logit") distribution family. For measures of pathogen load, and FOB, we used linear mixed effects models (*Harrison et al., 2018*). All models used the same repeated measures design. Treatment, sampling event, and their interaction were included as fixed effects in order to determine how each dependent variable was affected by our manipulation through time. Colony and bee yard were included as random effects in order to determine the among colony variance within each treatment and account for potential differences between bee yards. To examine how the *Varroa* load of migratory and stationary colonies differed over time with respect to FOB, we conducted

a separate linear mixed effects model. We first tested for temporal autocorrelation in the residuals of the model using an ACF plot and no autocorrelation was detected. For this model, we used FOB, treatment, time, and the resultant interactions as fixed effects and colony as a random effect. Significance for all models was determined using Type II Wald chi-square tests.

To examine potential differences in climate between California and North Carolina during the 27 days the migratory bees were in California, we used one-way Analysis of Variance (ANOVAs) on average daily temperature, precipitation, and wind speed by state (NOAA National Centers for Environmental Information).

We conducted all statistical analyses using the statistical software "R" (*R Core Team, 2016*). GLMMs were conducted using the lme4 package (v 1.1-13) (*Bates et al., 2015*). The corresponding Type II Wald chi-square tests were conducted using the Anova function in the car package (v 2.1-4) (*Fox & Weisberg, 2011*). Temporal autocorrelation was tested using the acf function. Classification analyses were conducted using the lda function in the mass package (v 7.3-45) (*Ripley & Venables, 2002*). The adonis function was used to perform PERMANOVA in the vegan package (v 2.4-3) (*Oksanen et al., 2014*).

## RESULTS

While in California, migratory colonies experienced similar weather conditions (mean daytime temperature, wind speed, and precipitation) to those experienced by stationary colonies in North Carolina ($F_{1,52} < 3.106$, $P > 0.084$; Table S1). All colonies were absent of IAPV. BQCV was present in all colonies for the duration of the study (Fig. S1).

### Experiment 1: migratory verses stationary

At the start of the experiment, there was no significant difference between migratory and stationary colonies in prevalence ($\chi_1^2 < 1.143$, $P > 0.285$) or load ($F_{1,30} < 3.01$, $P > 0.093$) of any of the four pathogens. In addition, there was no difference in FOB at the beginning of the experiment (migratory: $7.94 \pm 0.57$ sd, stationary: $7.44 \pm 0.51$ sd).

Upon the return of the migratory colonies, our pathogen and hive population measurements collectively predicted whether a colony was migratory or stationary (Fig. 2A). The linear combination (LD1) adequately discriminated between the migratory group and the stationary group and yielded correct classification rates of 87.5% for migratory colonies and 75% for stationary colonies. Also, prior to contact with the migratory colonies, the exposed colonies were similar to the isolated stationary colonies and essentially formed one large group (Fig. 2A). After contact with migratory colonies, there was statistically significant group separation between migratory and stationary treatments ($F_{1,30} = 5.03$, $P = 0.007$).

Migratory colonies returned from California with significantly higher BQCV loads compared to the stationary group ($\chi_1^2 = 16.488$, $P < 0.001$; Fig. 3A), and BQCV load increased with time (Fig. 3A and Table 1). The prevalence (Fig. S1) and load of DWV (Fig. 3B) did not differ between treatments following return of migratory colonies but both increased with time (Table 1). *Nosema* load and prevalence (Fig. S1) did not differ between treatments following return of migratory colonies and *Nosema* load decreased with
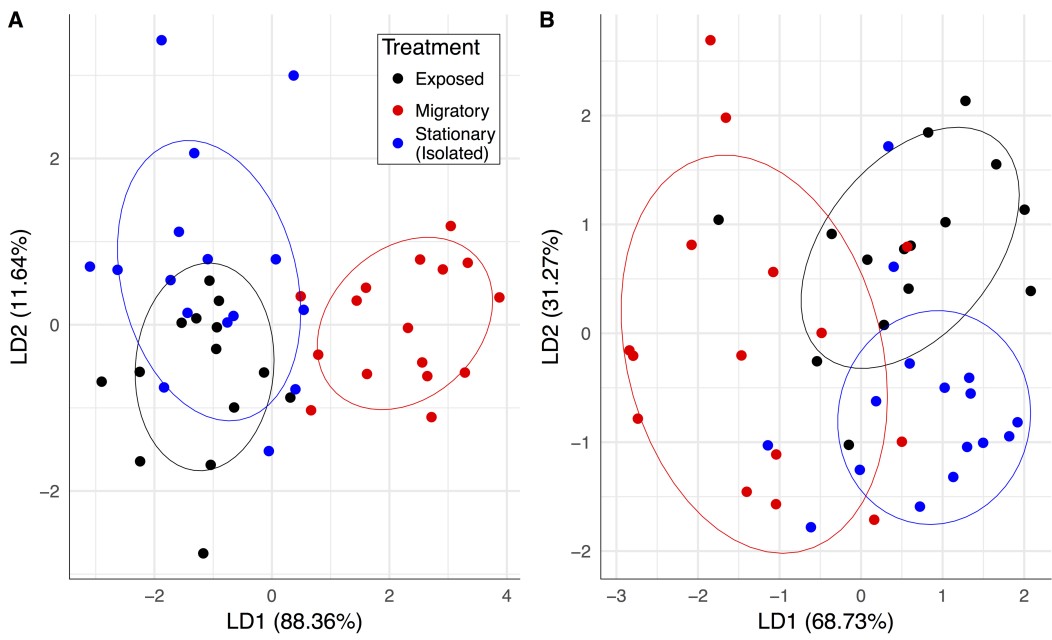

**Figure 2** **Pathogen community and colony health predicts treatment group membership.** Linear combinations from discriminant analyses created from all pathogen variables (except BQCV prevalence) and frames of bees for exposed (black), migratory (red) and stationary/isolated (blue) colonies. Axes represent the percentage of between group variance explained. (A) Experiment 1 at sampling event two, migratory and stationary colonies were separated by LD1 while stationary and exposed colonies are clustered. (B) Experiment 2 at sampling event three, after the exposed group had been allowed to forage alongside the migratory colonies, exposed and isolated were separated along LD2, while LD1 separated out migratory colonies. The significant PERMANOVA tests for both experiments corroborated the differences between group centroids. Circles represent 70% confidence intervals and are provided to visualize the centroids of each group.

time (Table 1). However, for *Varroa*, there was a significant treatment × time interaction (Fig. 3C). *Varroa* loads increased steadily for stationary colonies, but decreased in migratory colonies over the month after returning from California ($\chi^2_1 = 6.465, P = 0.011$). There was also a significant interaction of treatment × time for FOB, with migratory colonies returning with fewer FOB than their stationary counterparts ($\chi^2_1 = 5.651, P = 0.017$). There was a significant interaction of FOB × treatment × time on *Varroa* loads ($\chi^2_1 = 4.045, P = 0.044$) indicating that *Varroa* loads were differentially affected by FOB for each treatment group over time. Other interaction terms were not statistically significant (Table 1).

### Experiment 2: exposed verses isolated

At sampling event two, there was no significant difference between exposed and isolated stationary colonies in pathogen prevalence ($\chi^2_1 < 1.143, P < 0.285$) or load ($F_{1,30} < 1.279$, $P > 0.267$). FOB was similar between groups at the beginning of the experiment ($F_{1,29} = 0.858, P = 0.362$).

One month after the exposed group foraged alongside the migratory colonies, there was an increase in between-group separation with groups becoming more distinguishable from each other. While all groups separated in this third time step, the exposed and

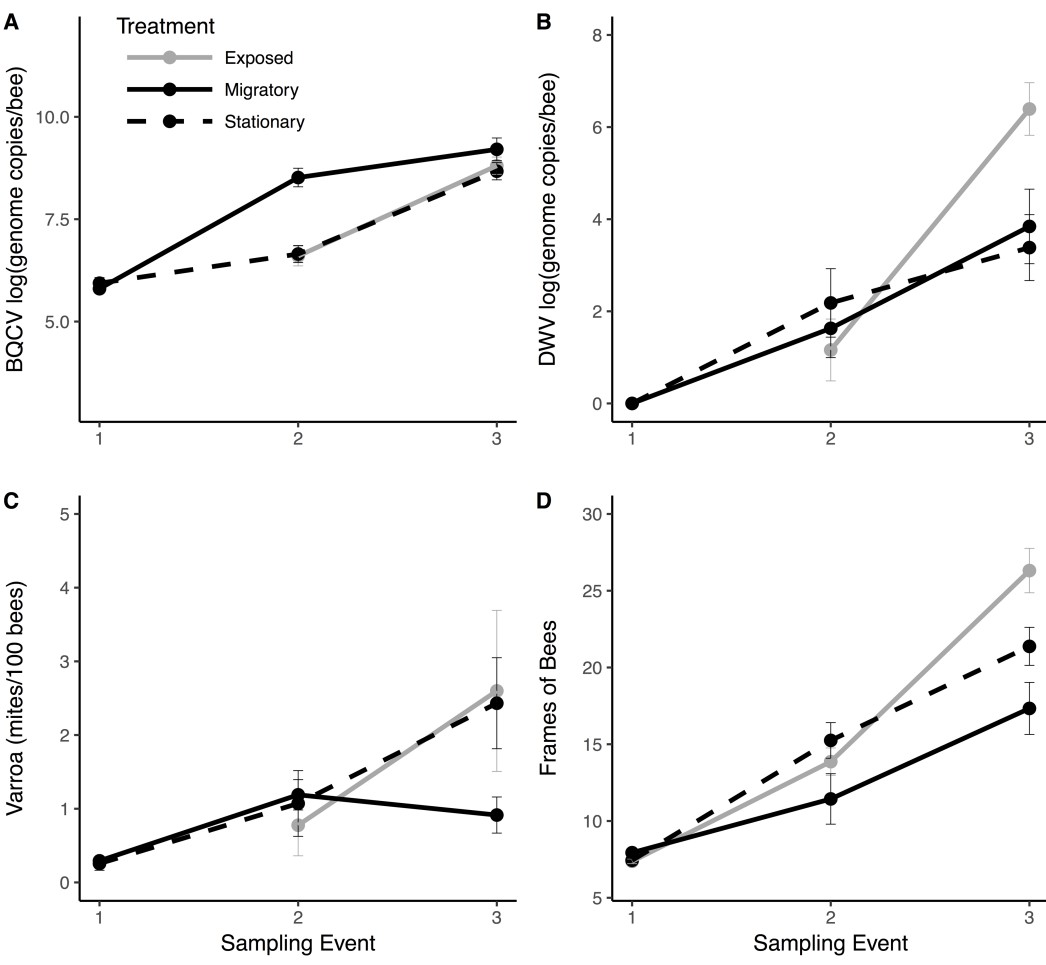

**Figure 3 Pathogen and colony population metrics for treatment groups through time.** Migratory (solid line) and stationary/isolated (dotted line) colonies were sampled at three time points and exposed (gray) colonies were sampled at two time points. Sampling event (1) occurred before migratory colonies were transported, (2) upon their return, and (3) one month after return. Panels show results for three pathogens and one health metric: (A) black queen cell virus (BQCV) in log genome copies per bee (B) deformed wing virus (DWV) in log genome copies per bee (C) *Varroa* load in mites per 100 bees and (D) Frames of bees (FOB), as a proxy for colony population. In Experiment 1: migratory verses stationary/isolated colonies, there was a significant effect of time for all measures. For BQCV, there was a significant effect of treatment. There was a significant time × treatment interaction for FOB and *Varroa*. In Experiment 2: exposed colonies verses stationary/isolated, there was a significant effect of time for each measure. For DWV, there was a significant time × treatment interaction. Bars represent standard errors.

migratory groups were less distinguishable from one another compared to the stationary group (Fig. 2B). The linear combinations (LD1 and LD2) yielded a correct classification rate of 75% for stationary colonies but correct classification rates for migratory and exposed colonies were lower, 43.75% and 56.25%, respectively. PERMANOVA results indicated statistically significant group separation between isolated, migratory and exposed treatments ($F_{2,43} = 4.72$, $P = 0.001$).

**Table 1** Summary statistics for experiment 1: migratory verses stationary.

| Variable | Effect | $\chi_1^2$ | P[a] | Sig[b] |
|---|---|---|---|---|
| DWV load | Treatment | 0.004 | 0.9512 | |
| | Time | 39.328 | <0.001 | *** |
| | Treatment:Time | 0.1592 | 0.690 | |
| DWV prev. | Treatment | 0.067 | 0.796 | |
| | Time | 15.805 | <0.001 | *** |
| | Treatment:Time | 0.024 | 0.878 | |
| BQCV load | Treatment | 16.488 | <0.001 | *** |
| | Time | 187.235 | <0.001 | *** |
| | Treatment:Time | 2.229 | 0.135 | |
| *Varroa* load | Treatment | 0.413 | 0.520 | |
| | Time | 18.391 | <0.001 | *** |
| | Treatment:Time | 6.465 | 0.011 | * |
| *Varroa* prev. | Treatment | 1.290 | 0.256 | |
| | Time | 4.896 | 0.0270 | * |
| | Treatment:Time | 3.21 | 0.073 | |
| *Nosema* load | Treatment | 0.645 | 0.422 | |
| | Time | 30.855 | <0.001 | *** |
| | Treatment:Time | 0.280 | 0.596 | |
| *Nosema* prev. | Treatment | 0.007 | 0.931 | |
| | Time | 3.652 | 0.056 | |
| | Treatment:Time | 3.352 | 0.067 | |
| FOB | Treatment | 3.597 | 0.058 | |
| | Time | 152.838 | <0.001 | *** |
| | Treatment:Time | 5.651 | 0.0174 | * |

**Notes.**

DWV load, deformed wing virus load; DWV prev., deformed wing virus prevalence; BQCV load, black queen cell virus load; Varroa prev., Varroa prevalence; Nosema prev., Nosema prevalence; FOB, frames of bees.

Prevalence is the percentage of colonies positive for a pathogen (DWV, Nosema, and Varroa). Virus load (DWV and BQCV) results for each colony are presented in average virus genome copies/bee. Nosema load is reported as average number of spores/bee and Varroa is reported as number of mites per 100 bees.

[a] Significance for all models was determined using Type II Wald chi-square tests.

[b] Asterisks represent level of significance.

We found no effects of treatment (exposed verses isolated) for any of the parasite or disease response variables (Fig. 3). However, *Varroa* prevalence and load, *Nosema* prevalence and load, and BQCV significantly increased with time (Table 2). There was a significant treatment × time interaction for both DWV load ($\chi_1^2 = 9.229$, $P = 0.002$; Fig. 3B) and DWV prevalence ($\chi_1^2 = 4.94$, $P = 0.026$; Fig. S1) such that DWV in exposed colonies increased at significantly higher rates than the isolated group. There was also a significant treatment × time interaction for FOB ($\chi_1^2 = 9.946$, $P = 0.0016$; Fig. 3D) with exposed bees increasing at a significantly higher rate compared to the isolated group. Other interaction terms were not significant (Table 2).

**Table 2   Summary statistics for experiment 2: exposed verses isolated.**

| Variable | Effect | $\chi^2_1$ | P[a] | Sig.[b] |
|---|---|---|---|---|
| DWV load | Treatment | 2.056 | 0.152 | |
| | Time | 23.510 | <0.001 | *** |
| | Treatment:Time | 9.229 | 0.002 | ** |
| DWV prev. | Treatment | 0.025 | 0.874 | |
| | Time | 8.811 | 0.003 | ** |
| | Treatment:Time | 4.945 | 0.026 | * |
| BQCV load | Treatment | 1.355 | 0.244 | |
| | Time | 58.001 | <0.001 | *** |
| | Treatment:Time | 0.054 | 0.816 | |
| *Varroa* load | Treatment | 0.471 | 0.493 | |
| | Time | 23.658 | <0.001 | *** |
| | Treatment:Time | 0.191 | 0.662 | |
| *Varroa* prev. | Treatment | 1.390 | 0.238 | |
| | Time | 10.129 | 0.001 | ** |
| | Treatment:Time | 0.060 | 0.806 | |
| *Nosema* load | Treatment | 0.882 | 0.348 | |
| | Time | 37.926 | <0.001 | *** |
| | Treatment:Time | 0.260 | 0.610 | |
| *Nosema* prev. | Treatment | 0 | 0.1 | |
| | Time | 7.771 | 0.005 | ** |
| | Treatment:Time | 0.004 | 0.950 | |
| FOB | Treatment | 1.899 | 0.168 | |
| | Time | 89.191 | <0.001 | *** |
| | Treatment:Time | 9.946 | 0.002 | ** |

**Notes.**

DWV load, deformed wing virus load; DWV prev., deformed wing virus prevalence; BQCV load, black queen cell virus load; Varroa prev., Varroa prevalence; Nosema prev., Nosema prevalence; FOB, frames of bees.

Prevalence is the percentage of colonies positive for a pathogen (DWV, Nosema, and Varroa). Virus load (DWV and BQCV) results for each colony are presented in average virus genome copies/bee. Nosema load is reported as average number of spores/bee and Varroa is reported as number of mites per 100 bees.

[a] Significance for all models was determined using Type II Wald chi-square tests.

[b] Asterisks represent level of significance.

# DISCUSSION

Migratory pollination services are an essential component of the US agricultural economy, yet this practice exposes honey bee colonies to a combination of factors that may compromise individual bee and colony health. Although there is widespread concern that migratory pollination can place honey bee colonies at increased risk to acquire and spread pathogens and parasites, there is a lack of experimental evidence demonstrating this phenomenon. Here, we controlled for management practices and starting conditions as well as the time at which bees were sampled for diseases and parasites. Our results show that while migratory conditions can negatively affect colony health and increase disease load, in some cases these impacts were transient.

With the exception of *Nosema,* honey bee colonies experienced an increase in pathogen prevalence and load over time with the highest levels occurring during the last sampling

event in March, following the seasonal trends of other time-course studies (*Tentcheva et al., 2004*; *Runckel et al., 2011*). Peak incidences of these viruses occur in warmer months when transmission is more likely to occur as a result of increased brood rearing (*Chen & Siede, 2007*) and increased foraging (*Singh et al., 2010*) However, for BQCV and *Varroa*, our results indicate that bees in the migratory conditions were affected differently compared to their stationary counterparts.

The migratory colonies in our study returned from almond pollination with higher BQCV loads compared to the stationary colonies but had converged to similar levels one month later indicating that migratory conditions exacerbated BQCV infection but these effects were transient. Colonies experience stress during transportation (*Simone-Finstrom et al., 2016*) which impairs immunity (*James & Xu, 2012*) and promotes elevated levels of virus replication. Pollinators of large monocultures experience a reduction in forage diversity (*Decourtye, Mader & Desneux, 2010*; *Colwell et al., 2017*) which increases susceptibility to disease (*Di Pasquale et al., 2013*). Exposure to agricultural chemicals adversely affects the insect immune response and promotes replication of RNA viruses in bees (*Di Prisco et al., 2013*; *Doublet et al., 2015*). In particular, higher BQCV titers are associated with exposure to organosilicone surfactant adjuvants (OSS), a class of surfactants used to enhance the spread of the active ingredient (*Fine, Cox-Foster & Mullin, 2017*). OSSs are heavily used in California almonds during the late January to March bloom period when migratory colonies are present (*Ciarlo et al., 2012*; *Mullin et al., 2016*). In addition to OSSs, bees involved in almond pollination may also be exposed to a wide range of pesticides. In recent years, the use of insecticides, herbicides, and fungicides has steadily increased in California almond crops (*CDPR (California Department of Pesticide Regulation) CalPIP, 2016*). We did not measure pesticide or OSS exposure in our colonies and are therefore cautious to speculate its role in the increased virus loads in our study. However, in light of our results and previous work, we believe pesticide-pathogen interactions in migratory colonies warrants further study.

Compared to stationary colonies, the migratory colonies had fewer FOB upon return from California. The lower population size observed may be a result of forager die-off after the large pollination event, as migratory bees have significantly shorter lifespans when compared to stationary bees as a result of increased oxidative stress (*Simone-Finstrom et al., 2016*). In addition, foragers could have been displaced during transit. As typical with migratory colonies, our colonies were moved to holding yards before and after pollinating almonds. When colonies are moved, foragers are forced to re-assess and re-learn their surroundings which can cause significant loss and/or drifting of foragers (*Nelson & Jay, 1989*). Despite migratory colonies returning with fewer numbers and remaining lower in FOB compared to stationary colonies, the two groups experienced similar population growth rates during the month following the large pollination event.

Upon return from California, mite prevalence and load in migratory colonies were similar to their stationary counterparts. However, when sampled one month later, mite prevalence and load in the stationary colonies had significantly increased, while mite prevalence and load in the migratory colonies declined slightly, and was significantly lower than that in stationary colonies. Since female mites must reproduce within the pupal cells
of developing honey bees, mite population growth is largely dependent on the availability of bee brood. Although we did not measure brood size, adult bee population size is highly correlated with brood size of the previous time step (*Torres, Ricoy & Roybal, 2015*) and mite population size (*Martin, 1998*; *DeGrandi-Hoffman et al., 2016*). Thus, the lower mite prevalence and load in migratory bees is likely, in part, a reflection of the lower reproduction of these colonies. Additional unknown factors may be influencing the lower mite loads in migratory colonies, as *Varroa* loads of the migratory and stationary colonies showed different relationships with FOB over time. Results of the US National Honey Bee Disease survey suggested that migratory beekeepers may treat with acaricide more effectively and the mechanical motion of the truck during transportation helps to dislodge mites from bees (*Traynor et al., 2016b*). Since our colonies returned from California with similar mite prevalence and load as the stationary group, it is unlikely that the motion of the truck had an impact. Additionally, we are confident that the difference in mites we saw during the last sampling event was not due to beekeeper practices as mite treatments were standardized across all groups.

Colonies exposed to migratory bees experienced a significantly greater increase in DWV prevalence and load compared to isolated colonies one month after foraging alongside the migratory colonies. *Varroa* loads could not explain this difference since exposed and isolated colonies experienced similar *Varroa* loads throughout the study. The greater population size of the exposed colonies in the last sampling event, could have increased dissemination of DWV. However, isolated colonies had higher bee populations than the migratory colonies and we saw no differences in DWV prevalence or load between those two groups. Previous studies found that DWV was a good predictor of weaker colonies (*Budge et al., 2015*) and thus one would not expect our results to simply be attributed to an increase in numbers and thus exposure. One potential explanation is that the migratory bees returned from pollinating almonds with a more virulent DWV strain that disseminated quickly in the exposed group as a result of their larger colony size and higher *Varroa* population (*Martin, 2002*; *Rosenkranz, Aumeier & Ziegelmann, 2010*; *Glenny et al., 2017*). Using deep sequencing, viruses not previously found in honey bees have been detected in migratory hives (*Runckel et al., 2011*) and recently, a more virulent recombinant of DWV was found to replicate at high levels when transmitted by *Varroa* mites (*Ryabov et al., 2014*). Despite this evidence, we remain cautious of speculating transmission of a novel or more virulent strain.

## CONCLUSIONS

Migratory bees are subjected to a myriad of stressors not experienced by their stationary counterparts including transport, lower diversity of floral resources, exposure to bees from tens of thousands of other colonies that may be diseased, and exposure to large quantities of pesticides. The migratory conditions in our experiment encompassed all these components, and thus we cannot attribute our results to a single or even an exact combination of causes. Furthermore, our study, while novel in scope, was conducted over a relatively short time span using a single set of migratory conditions and focused on a limited set of bee pathogens.

Thus, we are cautious to claim that our results are representative of migratory beekeeping, at large, but do suggest that migratory conditions may exacerbate BQCV infections and lead to slower colony growth. Future studies to examine the underlying mechanisms, individually and in concert, as well as those that encompass colony health and additional pest and pathogens over a longer time span will provide further insight.

A growing body of evidence suggests that pests and pathogens from managed bees are spilling over into wild bee populations (*Colla et al., 2006*; *Spiewok & Neumann, 2006*; *Hoffmann, Pettis & Neumann, 2008*; *Otterstatter & Thomson, 2008*; *Singh et al., 2010*; *Graystock et al., 2013*; *Levitt et al., 2013*; *Brown, 2017*). Sympatric bumble bees and honey bees are infected by the same strains of DWV (*Fürst et al., 2014*) and virus prevalence in honey bees is a significant predictor of virus prevalence in bumble bees (*McMahon et al., 2015*). The higher BQCV load we document in migratory bees could thus pose a risk to wild bees. It is also possible that increased disease load as a consequence of migratory pollination could affect honey bees in future years due to disease spillback from infected wild bees (*Graystock, Goulson & Hughes, 2015*). Therefore, it is important to test whether wild bee populations have higher disease prevalence in proximity to honey bee apiaries, particularly those with migratory management practices.

According to recent forecasts, the US demand for commercial crop pollination services is expected to rise, particularly for almond (*USDA National Agricultural Statistics Service, 2017c*). Thus, understanding the effects of this current model of crop pollination on bees and identifying where, when, and how to mitigate those effects are critical to the apiculture industry. Our work suggests that some effects, while important, may be transitory. Thus, honey bees may be resilient to some stressors imposed by migratory conditions and recuperation after large pollination events is important to maintaining healthy migratory colonies.

## ACKNOWLEDGEMENTS

We would like to thank Joseph Schall, Greer Sargeant and Kendall Eppley for their assistance in the laboratory, and Jeff Lee for transporting the honey bee colonies across the country to California and back.

### Funding

This work was supported through http://experiment.com crowdfunding (http://www.experiment.com/beekeeping) with contributions from: the Vermont Beekeepers Associations, NH Beekeepers Association, Aaron J. Schwartz, Karla Eisen and George H. Wilson, Phyllis E. Burnham, Kay Newhouse, Edward and Jacqueline Alger, Dan Boisvert, Allison Malloy, Jonathan Alger, Amy Handy, William Castro (http://beefriendlyapiary.com), Kay Newhouse, Karla Eisen and George H. Wilson, Caydee Savinelli, and Richard Reid. Partial funding for open access provided by the UMD Libraries' Open Access Publishing Fund. The funders had no role in study design, data collection and analysis, decision to publish, or preparation of the manuscript.

### Grant Disclosures
The following grant information was disclosed by the authors:
Vermont Beekeepers Associations.
NH Beekeepers Association.
UMD Libraries' Open Access Publishing Fund.

### Competing Interests
The authors declare there are no competing interests.

### Author Contributions
- Samantha A. Alger conceived and designed the experiments, performed the experiments, analyzed the data, contributed reagents/materials/analysis tools, prepared figures and/or tables, authored or reviewed drafts of the paper, approved the final draft.
- P. Alexander Burnham conceived and designed the experiments, performed the experiments, analyzed the data, contributed reagents/materials/analysis tools, prepared figures and/or tables, authored or reviewed drafts of the paper.
- Zachary S. Lamas conceived and designed the experiments, performed the experiments, contributed reagents/materials/analysis tools, authored or reviewed drafts of the paper.
- Alison K. Brody contributed reagents/materials/analysis tools, authored or reviewed drafts of the paper.
- Leif L. Richardson conceived and designed the experiments, contributed reagents/materials/analysis tools, authored or reviewed drafts of the paper.

### Data Availability
 GitHub: https://github.com/samanthaannalger/AlgerProjects/tree/master/Migratory Stationary.

### Supplemental Information
Supplemental information for this article can be found online at http://dx.doi.org/10.7717/ peerj.5812#supplemental-information.

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
