# Peer review of "Home sick: impacts of migratory beekeeping on honey bee (Apis mellifera) pests, pathogens, and colony size"

_PeerJ, doi:10.7717/peerj.5812_

## Round 0.1 · original submission · Major Revisions

Dear Dr. Alger and colleagues:

Thanks for submitting your manuscript to PeerJ. I have now received three independent reviews of your work, and as you will see, the reviewers raised many concerns about the research. Fortunately, these concerns are complemented with many praises, and overall I believe that a major revision that addresses these valuable critiques will greatly improve your manuscript. Should you choose to revise your work, good luck and please remember to address all of the concerns from each reviewer. Please do not overlook the edited manuscript file provided by reviewer 1.

I look forward to seeing your revision, and thanks again for submitting your work to PeerJ.

-joe

Reviewer 1 ·

Basic reporting

The authors have developed a commendable approach in an attempt to explain the effects of modern apiculture practices on honey bee colony health. Specifically, the authors attempt to explain the effects of a.) migration on colony population size and pathogen profiles, and b.) the effects of exposing stationary colonies to migratory colonies returning from California on pathogen profiles. The experimental approach developed in this study has the potential to resolve the complexity associated with modern apiculture on colony population health.

The introduction is clearly worded, and expertly organized. There is a clear conclusion paragraph which describes the goal of the study, and the main hypotheses being tested. However, the intro would benefit by elaborating on some mechanisms of colony health decline, and including relevant citations. For example, the species of Nosema responsible for colony health decline should be elaborated-on, considering evidence exists that Nosema ceranae has been linked with colony death in Spain (Higes et al. 2008).

Experimental design

The experimental approach is the main strength of this study. The authors are correct that many studies monitoring colony health lack an appropriate control group, and therefore I think this study will progress the field. However, I would like the authors to acknowledge a few experimental shortcomings:

a.) after returning from California, migratory colonies were placed next to stationary colonies in the same bee yard, but no additional colonies were placed next to the stationary colony. How can the authors conclude that pathogen exchange is as a result of migratory colonies, and not simply from increasing densities of bees in the yard? This relates to discussion lines 354-356

b.) if colonies are blocked spatially for the duration of the study, is it still appropriate to treat each colony as an independent individual? Are there block effects on colony condition?

c.) The analysis was performed on a limited set of bee pathogens. What might be the case if the authors were to survey for some more common bee pathogens (Lotmaria passim, LSV complex, KBV, PCR amplified Nosema ceranae, etc...)?

d.) The temporal scope of their study is somewhat short.

The statistical analysis is creative. Their use of multivariate statistics is important, and is a powerful tool to explain a complex system. However, I feel as if the authors can include more detail into the section regarding the building of linear mixed effects models, and the specific question they intended to answer through this analysis.

e.) was BQCV excluded from the discriminant analysis because it was prevalent in 100% of colonies? I would imagine this makes colonies more similar, and more difficult for the classification tree to discriminate treatment groups.

f.) There was no mention of including interactions in the models. Why include interactions?

g.) How did the authors deal with potential multicolinearity among fixed effects in their models? I would imagine there is temporal autocorrelation among FOB and mite loads.

Validity of the findings

The conclusions of the results are supported by current research in the field. The authors do an excellent job of stating the limitations of their limited temporal study in the conclusion paragraph. However, the authors sometimes do not explain results using honey bee biology.

a.) could the simplest explanation for FOB, mite, and pathogen increase in the third sampling event be because the third sampling event took place during the colony buildup phase? We would expect colony populations to grow to prepare for summer foraging. This could be why the authors find a lot of significant interactions between time of sampling and other fixed effects in the models.

b.) Lines 381-391, while interesting, seems beyond the scope of this study.

Annotated reviews are not available for download in order to protect the identity of reviewers who chose to remain anonymous.

Reviewer 2 ·

Basic reporting

Background Research in general ok, but referencing could be improved in some places.

Experimental design

The scope of the experiment appears quite limited in terms of number of colonies, exposure time, and observation time.

Validity of the findings

The discussion is not adequate in several places and in need of major improvement. Speculation is abundant.

Additional comments

L52 - "for over a decade honey bees have experienced catastrophic colony failure " is exaggerating and refers, if anything, to the US situation, but is not valid on a worldwide scale.
L 54 - the Zhu et al. 2014 paper deals with honey bee larvae in individual assays and is not a suitable reference for pesticide effects on the colony Level
L55 Varroa mites should be mentioned together with pathogens. I'm also not sure that Evans and Schwarz 2011 is the best refrence in this context; vanEngelsdorp and Meixner 2010 could be a better reference
L 57 - here, a more recent reference might be more suitable (p.ex. Traynor et al. 2016)
What about economy as driver for pollination or other apicultural practices?

The scope of the experiment appears quite limited in terms of number of colonies, exposure time, and observation time.
Sampling - at least for targeting Nosema infection levels, sampling away from the broodnest would have been more advisable (see Fries et al. 2013 JAR 52(1) (BEEBOOK)

Why was the number of brood cells assessed only once at the start of the experiment, and not again in the course of the experiment? Especially in the context with development of the Varroa mite population it would have been helpful to also obtain data on broodnest size.

Discussion:
L312 ff - why is it likely that migratory conditions exacerbated BQCV infection? The evidence is correlative, not causative. Is there any experimental evidence that migration indeed increases BQCV replication?

L317 ff, 320 fff – this reasoning is highly speculative, following from the correlative evidence above. You have no data that would show exposure of your hives to OSS, or allow the conclusion that this (putative) exposure would have resulted in higher virus replication. In addition, you would also have to show that the stationary colonies were not exposed to OSS.
Also - in line of this reasoning - how would you explain that one month later, the BQCV levels of all three groups (including the isolated and the exposed colonies) converged at an even higher Level?
In addition - BQCV infection levels in all three groups appear absurdly high (Fig. 3), compared to the published literature (where levels beyond 10^8 or 10^10 are considered as "high"

L337 ff - as there are no data on the amount of bee brood present, it is difficult to compare the mite infestation levels on adult bees of the three groups. Mite population dynamics and the ratio of mites in brood to mites on bees is highly dependent on the amount of brood present, but can also be influenced by contact with highly infested neighboring colonies (drifting, robbing etc.)

L360 ff - I'm not sure the higher DWV loads in the exposed colonies compared to migratory and isolated colonies can be explained from your data at all .
If I were to speculate, I might say that it looks like these colonies grew faster (they had more bees), but probably they also had a higher brood infestation level (because they probably also had more brood, but this wasn't measured). More mites in the brood usually also results in higher DWV replication in the brood, and consequently in higher DWV levels

Reviewer 3 ·

Basic reporting

The paper is generally clearly written and nicely organized. Although some sentences could benefit of clarity by being much concise or rephrase, the manuscript was easy to read. The introduction and background are highlighting the need to better understand the link between parasitism / pathogens risk associated with migration in honey bees. The work offered to test and quantify the effect of migration on colony size, pathogens prevalence and loads among 48 colonies submitted to different risk exposure despite originated from the same apiary. The results of different tests converged toward a link between beekeeping migration for pollination and negative impact on colony health in honey bees.

I would suggest a few modifications/check in the introduction part to ease the reader in the topic:
- L49: Please check the value you are stating from Gallai et al. 2009. I remember reading in this reference that insect pollination services, including bees, was estimated to worth about 153 billion and not 200 billion.

- I understand that you the aim is to highlight the management practice by showing its importance in US. I would suggest that one line would be added (maybe in the first paragraph L47-57) to indicate why pollen and nectar are important for honey bee colonies. As example you could indicate that pollen is essential for providing the brood the protein needs for good development.

- Since you are also assessing Varroa mites infestation here, I would suggest that you separate pathogens (Nosema, viruses and bacteria…) from parasite (Varroa destructor). This is much clearer for the naïve reader and accurate.

- L82: Where did you find this value for theoretical number of bees (colonies) sharing the same resource patch?

- L107: Here you are writing properly the name of the three viruses you tested but then in other section you directly write BQCV, IAPV and DWV. Remember that those acronyms are not general knowledge and please add abbreviation there.

Figures and table are good quality and generally well labelled. Some minor changes should be added.
Figure 1: Is it possible to add the geographical distance between your stationary and migratory colonies?
Figure 2: The blue label signification is indicated in the caption but since you made the effort to specify for the other treatments the legend in the figure, could you do the same for stationary colonies?

The paper generated data on health status of 48 Western honey bee colonies established in US before, during and after migration event. The raw data are all supplemented including weather measures. In your supplementary table 3, please indicate what is the value for each column. As an example I couldn’t understand what the value of BroodPattern meant. Follow the good example of your first supplementary table 1.

Experimental design

The experimental design is generally well organized but some details are difficult to understand. The number of colonies assessed here is well distributed among the treatments and reasonable for the research question. On a first sight, the three sampling points seems limited but given the period and the organism studied, it is more than reasonable (3 months, generation time honey bee 21 days). The methods used to detect pathogens and quantify parasite are following properly standards and seems reliable and replicable. Some minor changes, additions, modifications would be suggested as follows:

- L130: I do not completely understand how the measure of frame of bees worked. Were you checking the number of frame with bees on it (with or without brood) only? Did you also estimate the brood availability and eventually of drone brood too, given that you studied Varroa prevalence and load? Varroa population an so load is highly associated with brood availability (Beaurepaire et al. 2017).

- L134: Were all the queens A. m. carnica/A. m. ligustica genetic background check? How many colonies and apiaries are surrounding the stationary yard?

- L182: I would indicate here directly rather than in your results (cf L249) the absence of IAPV in your data. This will avoid your reader to wonder if you forgot to write the IAPV test in the material methods after line 182.

- Line 201: “We verified RT-PCR analysis through sequencing” – which sequencing? Could you specify the machine and length of sequences?

- Line 231 – 239: in the result section is visible that GLMMs also tested for interaction effects (mentioned in line 270-277). Please make this clear in the Method part. In addition, isn’t it necessary to correct p-values for multiple testing?

Validity of the findings

The results are reasonable given the experiments and authors are really cautious in their conclusion. PeerJ standards requirements are met in this study.

Additional comments

It is a really interesting paper and you have done a really good experimental job in assessing the impact of migration in honey bee colonies.

I wish you could have also take some pictures of your bee colonies (if you haven’t done it already) so you could show the reason your colony size is so different before and after treatment. It is not always clear for me if it is only the foragers missing or maybe a stress in brood production. If you have it will always be valuable in the supplementary data to put some pictures to show the evolution overtime of FOB.

In general, you did a good job discussing the different factors but you did not mention that monoculture has been shown in several publications to increase host susceptibility towards pathogens and toxins. Can easily be a part of the sentence of Line 315 – 319.

---

## Round 0.2 · Minor Revisions

Dear Dr. Alger and colleagues:

Thanks for re-submitting your manuscript to PeerJ. Unfortunately, none of the previous reviewers were available to re-review your revision. However, I was able to obtain a new review of your revision. This reviewer is very optimistic about your study, and has taken the time to point out some minor issues. Therefore, please address these new concerns in your revision. I believe that these final adjustments will make the work suitable for publication in PeerJ.

I look forward to seeing your revision, and thanks again for submitting your work to PeerJ.

-joe

Reviewer 4 ·

Basic reporting

The authors developed a design which makes it possible to explain the effects of our modern agricultural demand and pollination practices on the health of honey bees. Specifically, the effect of migration to and from almond monoculture pollination where bee colonies are facing tough conditions like poor diet, high pesticide loads and meeting thousands of bees from other locations, is studied. The strength of this study is the experimental design which is not only focusing on the bees exposed to the migration but also on the question if they will transmit disease to stationary colonies when shifted back home. This point is highly relevant in terms of disease spread among honey bee colonies over wide areas and has potential to answer many highly interesting questions in the future.
The introduction is clearly worded and well organized.

Line 72: There are several publications which clearly show that almost all monocultures are less good for honey bees than diverse diets are. Although, the authors give such citations in the discussion (Line 352/353), it would be nice to also include them here, otherwise the “may” does not sound like there are already published data on that. And the next sentence points clearly to a publication about travel stress. I would prefer if all stress factors we have data about are being pointed out like this in the introduction.

Line 76/77 A short sentence what the other publications found, would be great.

Line 86: which are not but

Line 100 to America and Europe

Experimental design

As already stated above, the experimental design with the proper and elegantly planned controls is the strength of this work.
The method part is well described and points like supplemental feeding practices etc are clearly reported and also methods used to detect and quantify pathogens and parasites are reliable and reproducible.

The temporal scope of the study, although being long enough for the questions the authors want to address, is relatively short and a later time point would have been interesting to study long-term effects, e.g. it would be highly interesting to know if the overwinter success of the different treatments differ.

The focus lays on a small set of bee pathogens. While it is obvious that a decision for some has to be made, it would have been a more spectacular result if the authors would have tested for pathogens previously not present in the colonies and being brought back by the migratory colonies. This would have been the next dimension of the work cited in 406/407. But I understand that there is always a time and budget limit.

While the methods are very strong in terms of sample size and experimental design of planning the three treatment groups, I wish the authors would have measured brood size and pollen load in the frames at each sampling time point. This would have given a better indication about potential reasons for the population size decline and if the Varroa decrease in the migratory colonies has something to do with lower brood availability. Anyway, a careful discussion about this problem is provided by the authors which is the most important point when realizing gaps in the design.

Validity of the findings

The conclusion and discussion are supported by current research in the field and the authors do a good job discussing the limitations of their design and are careful with speculation and if they do, they state it clearly.
I for example really liked the discussion point that potentially a brought back more virulent Virus strain could explain infection pattern – this could be an amazing study in the future.
I also enjoyed that the authors are mentioning potential impact on wild bee populations.

The authors point out correctly that they did not measure pesticides in the bee tissue or colony but they give a statement about OSS. As there are likely also other pesticides which have been used and can have an influence on the bee e.g. homing ability of the forager, I would suggest to list all of them in the supplemental material for detailed information if they have access to this information (e.g. from the almond industry?). Otherwise, it sounds a bit biased to just pick out one for making a speculative point.

Line 441 large pollination events or a large event

Figure 2: please state what the circles around the centroids indicate

Additional comments

Do you have potentially taken pictures of the frames which might make a later rough estimation of bee bread and brood load possible? That would be very valuable and potentially worth to try to implement. Unfortunately, the Varroa load has no real meaning, now as the reason cannot be explained.
I know, it is always easy to point something like this out afterwards. Otherwise, I really liked the well-planned experiment in general and I think that you can tackle many highly interesting questions with this design in future studies. Good luck!

---

## Round 0.3 · accepted · Accept

Dear Dr. Alger and colleagues:

Thanks for again re-submitting your manuscript to PeerJ, and for addressing the concerns raised by reviewer 4. I now believe that your manuscript is suitable for publication. Congratulations! I look forward to seeing this work in print, and I anticipate it being an important resource to the honey bee community. Thanks again for choosing PeerJ to publish such important work.

Best,

-joe

#